# The Behavioral Intention and Influencing Factors of Medical Staff toward COVID-19 Vaccinations

**DOI:** 10.3390/healthcare10040628

**Published:** 2022-03-27

**Authors:** Kuan-Chieh Tseng, Chao-Chien Chen, Jen-Min Huang, Tong-Hsien Chow, Chin-Hsien Hsu

**Affiliations:** 1MA Program in Social Enterprise and Cultural Innovation Studies, College of Humanities & Social Sciences, Providence University, Taichung 41170, Taiwan; jackt72@pu.edu.tw; 2Department of Leisure and Recreation Management, Asia University, Taichung 41354, Taiwan; peter72@asia.edu.tw; 3Department of Physical Education, National Pingtung University, Pingtung 90003, Taiwan; jenmin650317@gmail.com; 4Department of Leisure Sport and Health Management, St. John’s University, New Taipei 25135, Taiwan; thchow1122@mail.sju.edu.tw; 5Department of Leisure Industry Management, National Chin-Yi University of Technology, Taichung 41170, Taiwan

**Keywords:** COVID-19, COVID-19 vaccine, barrier, behavioral intention, vaccinations

## Abstract

This study mainly explored the behavioral intention and influencing factors of medical staff toward COVID-19 vaccinations. Medical staff were taken as the research subjects. This study selected 300 research subjects by the intentional sampling method and conducted a questionnaire survey. A total of 260 questionnaires were recovered (a recovery rate of 86%), and the number of valid questionnaires was 212, for an effective questionnaire rate of 81%. SPSS and AMOS were used for statistical analysis. As known from the research results: (1) medical staffs’ perception of COVID-19 vaccinations had a positive and significant impact on their behavioral intention for receiving COVID-19 vaccinations; (2) medical staffs’ perception of COVID-19 vaccinations had a negative and significant impact on the barriers to receiving COVID-19 vaccinations; (3) medical staffs’ motivation of receiving COVID-19 vaccinations had a positive and significant positive effect on their behavioral intention of receiving COVID-19 vaccinations; and (4) medical staffs’ motivation of receiving COVID-19 vaccinations had a positive and significant impact on the barrier to receiving COVID-19 vaccinations.

## 1. Introduction

The outbreak of COVID-19 in early 2020 has severely impacted a wide range of global social and economic aspects. According to the 2021 data from the Ministry of Health and Welfare’s Centers for Disease Control, as of 24 August 2021, there were 21,232,469 confirmed cases and 4,448,150 cumulative deaths throughout the world, and the global fatality rate was 2.09%. Among the above numbers, 15,392 confirmed cases and 828 deaths occurred in Taiwan. This article took the medical organizations of Taiwan as the research scope and the medical staff in Taiwan as the research subjects. The listed data were intended to explain that during the impact of this wave of the pandemic, while the load of Taiwan’s medical system was facing a serious test, the medical staff as the front-line personnel in the medical field were also faced with relatively high physical and psychological pressure. Therefore, the behavioral intention and influencing factors of medical staff being vaccinated against COVID-19 were further explored. The spread of the pandemic has not only caused dramatic changes in the medical system and daily medical business but also produced a heavy physical and psychological pressure on the general public. As pointed out in the research of Chien et al. [1], the COVID-19 pandemic requires social distancing in daily life. In hospitals, people’s appointments with their doctors, as well as their visits to relatives and friends, have been restricted, and the rules for both hospital personnel’s contacting patients and invasive treatments have also needed to change. The distance and isolation between people, and the anxiety and insecurity caused by the virus, have gradually grown and spread with the pandemic. However, front-line medical staff are the first to bear the brunt of the COVID-19 pandemic. Lo et al. [2] pointed out that the nursing industry is a profession with high work pressure that comes not only from the work itself but also from patients, family members, related health care personnel, and the hospital system. In the face of highly infectious diseases, the physical and mental health of nursing personnel are also affected. This view was verified by the study of Lancee et al. [3], which indicated that even one year after the end of the severe acute respiratory syndrome (SARS) epidemic, the lifetime prevalence of depression, melancholia, or drug abuse among medical personnel was about 30%. One of the subjects in that study even suffered from post-traumatic stress disorder. It can be seen that when medical staff directly care for patients, the time used for contacting patients is extremely long, the workload is heavy, and the risk of infection is high. In view of this, how to effectively prevent and protect front-line medical staff has become a focus of attention.

Today, the world is committed to promoting COVID-19 vaccination programs. The Central Epidemic Command Center under the Taiwan Centers for Disease Control (2021) pointed out that its COVID-19 vaccination plan for 2021 is to prevent people in the fields of medicine, epidemic prevention, and the maintenance of social operation and national security from being infected with COVID-19. The data were provided by the Centers for Disease Control, Ministry of Health and Welfare (Taiwan) in 2021. The data were used to illustrate the importance of vaccination for Taiwan’s medical staff and whether negative cases after vaccination would affect the medical staff’s motivation for vaccination. The reason behind this promotion is that their contraction of COVID-19 may turn them into the source of infection or affect their services in providing medical care, in epidemic prevention, in social operations, and in national security. The data cited by the author at the time were applicable to the current situation of the pandemic in 2021. However, the Centers for Disease Control, Ministry of Health and Welfare pointed out that, in 2020, “two doses of the COVID-19 vaccines can improve immunity and effectively prevent COVID-19 infections, as well as lower the risk of severe illness and death after infection. However, the protective efficacy of the vaccine will gradually decline after vaccination, thus, even if two doses of vaccination are completed, breakthrough infections may still occur due to the COVID-19 virus variant strain. As a result, for those who received two doses of adenovirus vector COVID-19 vaccines (including the COVID-19 vaccines of AstraZeneca and Janssen), the Moderna vaccine can be used as a booster dose. People who have completed two doses of vaccines and meet the requirements for vaccination are recommended to receive a third dose of the COVID-19 vaccine to increase the autoimmune protection, in order to reduce the risk of severe illness after infection”. The Centers for Disease Control of the Ministry of Health and Welfare (2021) also pointed out that, taking the AstraZeneca vaccine as an example, the vaccine does not contain replicable SARS-CoV-2 virus particles, and people will not suffer from COVID-19 due to this vaccine [4]. Completing two doses of vaccination can prevent 63% of the risk of symptomatic infection. According to the analysis of clinical trial data, when the vaccination interval is 12 weeks between doses, the protective power of the vaccination is about 81% (60–91%). The above data illustrate the effectiveness of the vaccine. Although the vaccine provides an extra protection for medical staff, with the increase in the adverse events after COVID-19 vaccination and serious adverse events after vaccination [5], it is worth further exploring whether the adverse events of vaccination have an impact on the willingness of medical staff to get vaccinated against COVID-19. In other words, further empirical research is needed on issues, such as the perception of COVID-19 vaccinations and whether the motivation to get vaccinated will affect the vaccination behavior, as well as the impacts of barriers on the vaccination motivation.

It can be seen from the relevant literature that research related to COVID-19 for medical staff has mostly focused on the psychological impact [6,7,8,9] as well as descriptions of the experience of medical staff in caring for infected people [6,10]. However, there are few studies on the behavioral intention of medical staff to get vaccinated against COVID-19. According to Ray [11], more than one billion people globally were reluctant to receive COVID-19 vaccinations. Even with free vaccinations, only 68% of adults agreed to be vaccinated, while 29% still said they would refuse it. It can be learned from the data that the reasons people hesitated to receive vaccinations were complicated. As medical staff are the front-line personnel in the medical field, it is important to understand their perception of and motivation for vaccinations. The main research purposes of this article are to understand the intentions of medical staff in receiving vaccinations through analyses, and to promote the relevant government units to formulate appropriate response strategies in real time and based on available evidence. As mentioned above, issues related to COVID-19 vaccinations, such as “vaccine hesitancy”, “vaccination side effects affecting work”, “vaccine communication”, and other topics, were seldom discussed in relevant literature. However, these issues may affect medical staff’s motivation and create barriers to receiving vaccinations. This study conducted empirical research on this topic, which is the difference between this article and previous studies. Therefore, this article explored the impact of medical staff’s perception of and motivation for COVID-19 vaccinations based on the behavioral intention of receiving COVID-19 vaccinations. It also analyzed the impact of medical staff’s perception of and motivation for COVID-19 vaccinations regarding the barriers to receiving COVID-19 vaccinations. This research selected 300 medical staff as the research sample by intentional sampling and conducted a questionnaire survey, and a total of 260 questionnaires were recovered. After invalid questionnaires were removed, the data were archived with SPSS 24.0 statistical software (IBM SPSS Statistics 20, Chicago, IL, USA), and AMOS 24.0 statistical software (IBM SPSS Statistics 20, Chicago, IL, USA) was used to analyze the correlation between variables. The results show that medical staff’s perception of and motivation for COVID-19 vaccinations had a positive impact on the behavioral intention to receive COVID-19 vaccinations, and the impact was significant, while medical staff’s perception of COVID-19 vaccinations had a negative and significant effect on the barriers to receiving COVID-19 vaccinations; moreover, the results show that the research hypothesis regarding the impact of medical staff’s motivation to receive COVID-19 vaccinations on the barriers to receiving COVID-19 vaccinations was not supported. With the exception of this research hypothesis, which is different from the conclusions of previous studies, the remaining research hypotheses can be compared with relevant literature; thus, more cases can be provided as a reference for future research regarding medical staff receiving vaccinations.

## 2. Literature Review

### 2.1. Vaccination

The development of COVID-19 vaccines and vaccination are the top priority of many countries in the fight against COVID-19 at present. The purpose is to prevent the spread of the pandemic, improve the overall protection of the vaccine, prevent the public from symptomatic infection, and make the human body generate stronger immunity. The Central Epidemic Command Center (2021) pointed out that vaccination is the most effective preventive intervention for infectious diseases, as well as one of the most urgent and necessary prevention strategies in Taiwan to protect the health of the people against COVID-19. In order to build up herd immunity against COVID-19, the procurement target is to cover 65% of the vaccination demand by all the people. The purpose of the information provided by the Central Epidemic Command Center in 2021 was intended to explain how Taiwan obtained vaccines at a time when vaccines were out of stock globally. Therefore, this article used this information to illustrate the relevant resources of vaccines for Taiwanese front-line medical staff. Currently, plans for international investment (participating in the COVAX mechanism led by the World Health Organization (WHO), Global Alliance for Vaccines and Immunisation (GAVI), Coalition for Epidemic Preparedness Innovations (CEPI), domestic production, and purchasing from manufacturers are being carried out simultaneously. Vaccines are supplied by advanced countries, the European Union, and those with Emergency Use Authorizations (EUA). Medical personnel have long been on the front line in medical care, and their need for vaccination is more urgent than that of the general public. Therefore, on 29 April 2021, the Centers for Disease Control of the Ministry of Health and Welfare announced that it would consider medical staff, central and local government epidemic prevention personnel, and the people who live with high-risk front-line staff as high-risk groups for infection. In order to build up the immunity protection of these objects as fast as possible, the plan for the vaccination of these groups would be implemented from May 3 at public expense. The purpose of citing data from the Centers for Disease Control, Ministry of Health and Welfare in 2021 was to bring about a broader discussion on the medical staff mentioned in this article. Moreover, the people working in clinics and other medical institutions can also be referred to as medical staff in a broader sense. All non-medical personnel in clinics and other medical institutions (including pharmacies, blood donation institutions, pathological institutions, and medical laboratories) who were not included in the first category of vaccinations were also considered groups at a high risk of infection, and these personnel and the people who lived together with them were also included as subjects of vaccination by public funds at this stage. Although the Centers for Disease Control of Ministry of Health and Welfare has vigorously promoted the vaccination program for medical staff, the actual participation of medical staff in the program is worthy of further investigation.

### 2.2. Perception

The so-called ‘perception’ refers to the beliefs, evaluations, or opinions that people hold about things. These beliefs and evaluations are based on tangible evidence that an individual’s perception is a fact at a particular moment [12]. In other words, people process or interpret the information obtained by the senses in order to give a specific meaning to the environment. Chung [13] divided perception into the narrow sense and broad sense. The narrow sense of perception is interpreted as recognizing or knowing, which belongs to the lowest level of intelligent activities, as one only needs to know that the information exists. Therefore, perception is a kind of awareness that directly discovers, rediscovers, or recognizes the information of various images. In a broad sense, perception refers to the role of all forms of recognition, including feeling, inference, perception, memory, planning, decision making, attention, problem solving, and communicating ideas. The definition of perception in this study was based on the narrow interpretation of Chung [13], which refers to medical staffs’ understanding of COVID-19 vaccinations and the level of awareness and views on COVID-19 vaccines. For example, medical staffs’ awareness of COVID-19 vaccines could be learned by their answer to the item “COVID-19 vaccines can completely prevent the contraction of COVID-19” in the questionnaire.

### 2.3. Motivation

The purpose of people engaging in something is often to seek various levels of satisfaction through the process of participation, so different motivations are also generated. Chang [14] put forward that motivation refers to an internal process that causes an individual’s activity to move toward a certain goal. In other words, when people have needs in their hearts, it will drive their behavior to produce further actions. Chang [14] further pointed out that from a psychological point of view, organisms have psychological and physiological needs that cause them to move toward a relevant goal or stimulus in the environment and inspire continuous activities to meet those needs, alleviate emotional anxiety, relieve stress, and achieve a balance between body and mind. In terms of the motivation for receiving COVID-19 vaccinations, Robson [15] stated that scientists had begun to study the psychological factors of people’s hesitancy toward vaccines long before the emergence of COVID-19 at the end of 2019. He further pointed out that the factors are: (1) Confidence: Do people have confidence in the efficacy and safety of the vaccine? Do they have confidence in the health agency or manufacturer providing the vaccine? Do they have confidence in the government’s policy decision to introduce a vaccine? (2) Complacency: Do people worry that the disease will pose a threat to their health? If people are satisfied with their current situation, they will not get vaccinated. (3) Calculation: People try to weigh the pros and cons of vaccination by collecting a lot of data. (4) Constraints or convenience: Are vaccines readily available? How convenient are vaccinations? Are there any constraints? (5) Collective responsibility: Are people willing to be vaccinated to protect others? It can be seen that the motivations of medical staff to get vaccinated against COVID-19 are different, and whether these different motivations form a barrier to their vaccination needs to be further verified.

### 2.4. Behavioral Intention

Behavioral intention explores the influencing factors of whether people will or will not engage in a certain behavior. Generally, the Theory of Planned Behavior is mostly used to explore people’s behavioral intentions. This article mentioned that “The Theory of Planned Behavior was derived from the Theory of Reasoned Action proposed by Fishbein and Ajzen” [16]. Based on the reviewer’s comments, the wording “derived from the Theory of Reasoned Action” has been modified to “an extension of the Theory of Reasoned Action”. Thus, the amended sentence is “The Theory of Planned Behavior (TPB) is an extension of the Theory of Reasoned Action (TRA)”. The basic assumption of the theory is that human behavior is rational and that individuals can fully control their own behavior by will. The theory states that attitude is an individual’s positive or negative feelings toward a specific behavior, as well as an individual’s beliefs about a particular behavior. Attitudes can be said to be formed after the individual’s evaluation of a particular behavior is conceptualized. Subjective norms are individuals’ perceived social pressure for engaging in a particular behavior. Perceived behavioral control is the degree to which individuals have control or mastery over their own feelings when engaging in a particular behavior, as predicted by past experience in performing the behavior. In practice, however, not all behaviors can be controlled by an individual’s will. Instead, they will be affected by external objective environment or resource constraints. Taking this study as an example, medical staff may not be vaccinated due to the insufficient number of vaccines or other considerations in the process of vaccination. Therefore, it was worth further exploring the vaccination behavior of medical staff.

### 2.5. Barriers

Barriers refer to the inhibition of an individual’s actions, and the scope of this influence includes all the internal relationships or external barriers causing the individual to change the original behavior [17]. Exploring the barriers means exploring the reason why people do not do something. Yang [18] pointed out that the earliest concept of a barrier was proposed by Lewin [19], who used the perspective of social psychology to explain that individual behaviors are affected by various inhibitory forces of internal and external environments. Inner environments include the mental state, attribution, or personality, while external environments are a variety of experiences other than oneself. In this study, the barrier to medical staffs’ vaccinations against COVID-19 could be psychological worries about the side effects of the vaccines, as the external environment of the media has reported an increase in the number of adverse vaccination effects, which are both factors that may create a barrier for medical staffs to get vaccinated against COVID-19. Further empirical research is needed to understand the barriers of vaccination.

### 2.6. Relevant Research on Perception and Motivation and Behavioral Intention and Barriers

In the relationship between the variables in this study, perception has an effect on barriers [20,21], and the effect of motivation on barriers has also been demonstrated [22]. In addition, perception has a positive effect on behavioral intention [23,24], while motivation has a positive effect on behavioral intention [25]. Accordingly, this study took medical staff as the research subjects and conducted an empirical analysis on the behavioral intention to get vaccinated against COVID-19.

## 3. Methods

### 3.1. Research Structure

The purpose of this study was to explore the behavioral intention and influencing factors of medical staff to get vaccinated against COVID-19. According to the research purpose and relevant literature, the research structure proposed was as shown in Figure 1. The six research hypotheses were proposed and defined as:
**Hypothesis** **1** **(H1).***The perception of COVID-19 vaccinations has a positive and significant impact on the behavioral intention of receiving COVID-19 vaccinations.*
**Hypothesis** **2** **(H2).***The perception of COVID-19 vaccinations has a negative and significant impact on the barriers to receiving COVID-19 vaccinations.*
**Hypothesis** **3** **(H3).***The motivation for receiving COVID-19 vaccinations has a positive and significant impact on the behavioral intention of receiving COVID-19 vaccinations.*
**Hypothesis** **4** **(H4).***The motivation for receiving COVID-19 vaccinations has a positive and significant impact on the barriers to receiving COVID-19 vaccinations.*
**Hypothesis** **5** **(H5).***There is a significant gender-based difference in the behavioral intentions of medical staff.*
**Hypothesis** **6** **(H6).***There is a significant gender-based difference in the barriers of medical staff.*

### 3.2. Research Subjects

The purpose of this study was to explore the behavioral intention and influencing factors of medical staff to get vaccinated against COVID-19. Medical staff were taken as the research subjects, of which 300 were selected as the research sample by means of intentional sampling to conduct a questionnaire survey. The questionnaires were distributed from 10 May to 15 July 2021. A total of 260 copies of the questionnaire were recovered (a recovery rate of 86%) and the number of valid questionnaires was 212, revealing a valid questionnaire rate of 81%.

### 3.3. Research Tools

The content of the questionnaire in this study was mainly compiled with reference to the relevant literature of Chuang [26] and Lin [27] and the modification method for questionnaires. The questionnaire was divided into five parts with a total of 37 items, including eight items on basic personal information, 10 items on perception, 10 items on motivations, 4 items on intentions, and 5 items on the barriers. In this study, a five-point Likert scale was adopted. Each item was rated from 1 to 5 points, with answers ranging from “strongly disagree” to “strongly agree”.

### 3.4. Data Processing and Analysis

In this study, after the invalid questionnaires were excluded, the statistical data from the valid questionnaires were archived with SPSS 24.0 statistical software, and the correlation between variables was analyzed with AMOS 24.0 statistical software.

## 4. Results Analysis

### 4.1. Sample Characteristics

In this study, medical staff were selected as research subjects, with 212 valid samples. In terms of gender, 131 were male, accounting for 61.8% of the valid samples, and 81 were female, accounting for 38.2% of the valid samples; in terms of age, there were a total of 70 people aged 31–40, accounting for 33% of the valid samples, and a total of 7 people aged 61 and above, accounting for 3.3% of the valid samples; in terms of job titles, there were 112 fire rescue technicians, accounting for 52.8% of the valid samples, and there were 2 pharmacists, accounting for 0.9% of the valid samples; in terms of the medicine category, there were 191 people from Western medicine, accounting for 90.1% of the valid samples, and 21 people from traditional Chinese medicine, accounting for 9.9% of the valid samples; in terms of educational level, the majority of the subjects (102 people) had a bachelor’s degree, accounting for 48.1% of the valid samples, and 17 people had an educational level of senior/vocational high school or below, accounting for 8.0% of the valid samples; in terms of marital status, there were a total of 121 married people, accounting for 57.1% of the valid samples, and 9 people of other statuses, accounting for 4.2% of the valid samples; in terms of current working department, most subjects (111 people) were in the fire department/team, accounting for 52.4% of the valid samples, and there was only 1 person in emergency medical services, accounting for 0.5% of the valid samples, as shown in Table 1.

### 4.2. Comparison of the Differences in the Behavioral Intentions and Barriers between Medical Staff of Different Genders

The results of the *t*-test test show that the *p* value of the behavioral intention was 0.37 > 0.05, and this result is not significant, which means that there was no significant difference in the behavioral intentions of medical workers of different genders; the *p* value of the barriers was 0.01 < 0.05, which is significant, meaning that there was a significant difference in the barriers of medical staff of different genders, as shown in Table 2.

### 4.3. Test Model Analysis

The reliability and validity of the questionnaire in this study were analyzed by CFA, while the questionnaire items were modified by the main modification indices (MI) [28]. Perceptions 1, 2, 3, 7, and 9 in the perception scale were deleted; motivations 1, 2, 3, 4, 5, and 6 in the motivation scale were deleted, and barriers 4 and 5 in the barrier scale were deleted.

#### 4.3.1. Verification of Convergent Validity

Bagozzi and Yi [29] pointed out that the convergent validity of a questionnaire can be measured by its composite reliability (C.R.) and average variance extracted (AVE). For the good convergent validity of a questionnaire, the C.R. value should be greater than 0.6 and the AVE value should be greater than 0.5 [30]. In this study, convergent validity tests for the constructs of perception, motivation, intention, and barriers were conducted, and the factor loadings of all constructs were between 0.60 and 0.93. The C.R. values were between 0.83 and 0.83. 0.92, and the AVE values were between 0.50 and 0.80, thus meeting the criteria for convergent validity suggested by Bagozzi and Yi [29], Fornell and Larcker [30], and Hair et al. [31], indicating that the questionnaire of this study had good convergent validity, as shown in Table 3, Table 4, Table 5 and Table 6.

#### 4.3.2. Structural Model Analysis

The overall model fit of this study was tested with reference to the structural model analysis of Hair et al. [31], which included seven indicators, namely, the chi-square value (χ^2^), the ratio of χ^2^ to the degrees of freedom, GFI, AGFI, RMSEA, CFI, and PCFI. Bagozzi and Yi [29] indicated that the smaller the ratio of χ^2^ to its degrees of freedom, the better. The corrected ratio in this study was 1.19. Hair et al. [31] stated that the closer the GFI and AGFI values to 1, the better. The corrected GFI and AGFI values in this study were 0.93 and 0.91, respectively; Browne and Cudeck [32] pointed out that an RMSEA value less than 0.08 is the best. The corrected RMSEA value of this study was 0.03. The standard value of CFI should be greater than 0.90, and the corrected CFI value of this study was 0.99. The PCFI should at least be greater than 0.50. The revised PCFI in this study was 0.82, indicating that the overall fit of this study was good and the indicators met the criteria, as shown in Table 7.

## 5. Discussion

As shown in Table 8, Hypothesis 1 was supported, as the results indicated the perception of COVID-19 vaccinations had a positive and significant impact on the behavioral intention of receiving COVID-19 vaccinations. The results of this study were the same as the research results of Cheng [24]. The possible reason is that medical staff are on the front line of medical services and have rich information on vaccinations, so they have a high acceptance of vaccinations. H2 was confirmed, as the results indicated the perception of COVID-19 vaccinations had a negative and significant impact on the barriers to receiving COVID-19 vaccinations. The results of this study were consistent with the research conclusions of Chiang et al. [33]. A possible reason is that medical staff have more information about vaccinations, so there are fewer factors hindering their intentions to receive vaccinations. H3 was confirmed, as the results indicated the motivation for receiving COVID-19 vaccinations had a positive and significant impact on the behavioral intention of receiving COVID-19 vaccinations. The results of this study were the same as those of Kuo [25]. A possible factor is that medical staff understand their own needs for job safety and self-protection because of their workplace, and therefore, show a higher intention to get vaccinated. H4 was confirmed, as the results indicated the motivation for receiving COVID-19 vaccinations had a positive and significant impact on the barriers to receiving COVID-19 vaccinations. The results of this study were different from the research results of Tai [22]. A possible reason is that the number of adverse side effects of vaccinations has increased recently, and such adverse events may be a hindrance to getting vaccinated, even when medical staff have the motivation to get vaccinated, as shown in Figure 2 and Table 7.

## 6. Conclusions and Suggestions

### 6.1. Conclusions

With the increase in the rate of COVID-19 vaccinations and the concerted efforts of the Taiwanese people and relevant government units, the current pandemic situation is demonstrating a slightly easing trend. However, how to coexist with the virus is a new challenge in the second half of the pandemic, as a huge social cost is needed to permanently eradicate the virus. In the process of fighting the virus in the future, medical staff will always be the most important people on the first line of defense. In the face of relevant issues such as the use of mixed vaccinations from different brands and whether to have a third dose, it is necessary to explore the behaviors and possible problems encountered by medical staff in administering vaccines. This article analyzed the behavioral intention and influencing factors of medical staff to vaccinate against COVID-19 through empirical analysis. The following conclusions were made by this study:(1)H1 was confirmed, that is, the perception of COVID-19 vaccinations had a positive and significant impact on the behavioral intention for receiving COVID-19 vaccinations.(2)H2 was confirmed, that is, the perception of COVID-19 vaccinations had a negative and significant impact on the barriers to receiving COVID-19 vaccinations.(3)H3 was confirmed, that is, the motivation for receiving COVID-19 vaccinations had a positive and significant impact on the behavioral intention of receiving COVID-19 vaccinations.(4)H4 was confirmed, that is, the motivation for receiving COVID-19 vaccinations had a positive and significant impact on the barriers to receiving COVID-19 vaccinations.(5)H5 was not supported, that is, the behavioral intentions of medical staff of different genders were not significantly different.(6)H6 was supported, that is, the barriers of the medical staff of different genders were significantly different.

### 6.2. Suggestions

Based on the research results, this study put forth a number of suggestions for reference:(1)For Nursing Staff

The results of this study showed that the perception of vaccines had a negative and significant impact on the barriers. Therefore, it is suggested that medical staff improve their own absorption of vaccination-related information. For example, they should have a deep understanding of the policies for mixed vaccinations from different brands. Especially for the Omicron mutation found in South Africa that is gradually spreading, it is recommended that medical staff who are on the front line of medical treatments should take the initiative to understand the possibility of strengthening their self-protection by getting vaccinated with different brands of vaccines or by administering a third dose. At the same time, they should understand the registration methods and project records of the county and city health bureaus to which they belong, such as when the second dose can be administered, the status of mixed vaccinations using different brands, and the data on the enhanced protection provided by the third dose. By doing so, they may improve their own perception of vaccinations and reduce the barriers to vaccination. In addition, the results of this study showed that perception had a positive and significant impact on behavioral intention. It is, therefore, recommended that medical staff also understand the side effects of various brands of vaccines after administration, such as blood clots, fever, and strong side effects. They should also have relevant information on how to reduce side effects and further evaluate whether these side effects will affect their schedule or job duties, especially as they have more opportunities to access more updated vaccine information on the medical front line. Worrying about side effects is not a sufficient reason for refusing a vaccination, because the troubles caused by an unfortunate infection in the future could be far greater than these concerns. On the other hand, it is recommended that on-the-job education is implemented for medical staff to improve their perception of the seriousness of COVID-19. Meanwhile, they should also be familiar with standard operation procedures on medical sites, such as the standard operation of wearing and removing protective clothing, using the nursing technology in COVID-19 isolation areas, or the standard procedures on caring for COVID-19 patients in the hospital. Only through the self-awareness of COVID-19 and the implementation of standard operation procedures at the medical care site can medical staff have specific actions for COVID-19 and increase their intention to get vaccinated.

(2)For Medical Units

The results of this study showed that motivation had a positive and significant impact on behavioral intention. Therefore, from the perspective of medical unit managers, it is recommended to build a work environment that puts medical staff at ease, enhances their work motivation, and allows them to increase their intention to administer vaccines to jointly build a safer medical environment. Specifically, education and training may be offered to help medical staff overcome their fears of the COVID-19 pandemic. For example, as the Omicron variant from South Africa may soon become the global mainstream strain of the COVID-19 pandemic, medical unit managers need to be careful about how to respond to changes in nursing routines, as well as how to deal with different care environments from the past. Medical unit managers can establish correct response methods for medical staff through education and training, and at the same time reduce their negative emotions and establish a positive working environment. On the other hand, it is recommended that medical unit managers not only build a supportive working environment but also move toward adding equipment and devices related to epidemic prevention hardware, such as negative pressure aspirators, UV sterilizers, and personal protective equipment. In addition, specific actions can be carried out in a number of directions. For example, the space planning for medical staff and patients should be in separate chambers with various flow directions, the disinfection frequency of related equipment that is in high contact with patients should be fully adjusted, an epidemic prevention movement diagram for special wards should be created, a COVID-19 disease teaching plan should be formulated, standard operating procedures and nursing checklists should be developed, and hand hygiene and health control should be implemented. Moreover, it is recommended that medical unit managers establish a material management area on the information platform of the medical unit to allocate, supplement, check, and record epidemic prevention materials every day. They should also monitor the applicability of the relevant epidemic prevention materials to effectively manage the epidemic prevention materials. This will give medical staff confidence in the epidemic prevention capability of their working environment and further strengthen their motivation to continue engaging in medical work.

(3)Future Research

This article explored the vaccination behaviors and barriers of nursing staff through quantitative empirical research. In the future, it is recommended to further explore the emotional labor issues of medical staff through in-depth interviews. Understanding these emotional issues could facilitate an in-depth interpretation of the vaccination and work intention of medical staff, and even identify the psychological issues of medical staff caring for infected patients. For example, medical staff must face the pressure of caring for confirmed patients. This pressure may come from a lack of nursing experience or a fear of being infected. The spread of the pandemic has led to the shortage of medical manpower and caused medical staff to have different emotions about nursing ratios, use of auxiliary manpower, rotation schedule calculations, and rest. These emotions may affect their willingness to remain working in a medical environment. In addition, through in-depth interviews, nurses could have the opportunity to express their safety concerns in the medical field. It is not easy to present these psychological issues through quantitative research, so it is suggested that future research can be carried out in this direction.

In terms of the application of the Theory of Planned Behavior, this study only explored the behavioral intention and influencing factors of medical staff to vaccinate against COVID-19, but failed to further understand the relationship between the behavioral intentions and behaviors. Although behavioral intentions can effectively explain actual behaviors, as mentioned above, the factors that influence vaccination intention are very complex. Thus, future research should conduct an in-depth exploration of the actual behavior to improve the application value of the research results.

Regarding the limitations of this study, the research subjects of this study were limited to medical staff in Taiwan. However, the regulations and restrictions on vaccinations of front-line medical staff vary from country to country; therefore, there are limitations to the inference of the conclusions of this study. Moreover, the focuses of medical staff in different countries regarding vaccination are inconsistent; for example, the study of Khamis, Badahdah, Al Mahyijari, Al Lawati, Al Noamani, Al Salmi, and Al Bahrani in 2021 on Oman medical staffs’ attitudes towards COVID-19 vaccines noted that the concerns stated by the participants about the COVID-19 vaccines included unknown health issues, efficacy, and safety. The study also noted that physicians and male medical staff in Oman had more positive attitudes toward the COVID-19 vaccines than nurses and female medical staff; therefore, the inference is limited. On the other hand, there is still room for improvement in the number of samples in this study; therefore, in view of this research limitation, it is suggested that the questionnaire in future related research can be designed according to the norms of vaccination in various countries. In addition, it is recommended to obtain a larger sample size to better represent the population, meaning the results obtained through the analysis of a larger sample size will be more applicable to the population.

## Figures and Tables

**Figure 1 healthcare-10-00628-f001:**
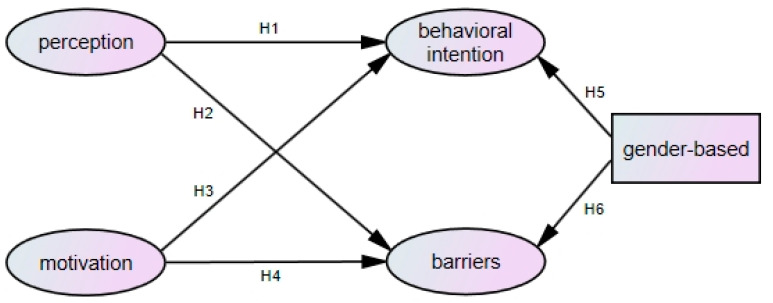
Research structure.

**Figure 2 healthcare-10-00628-f002:**
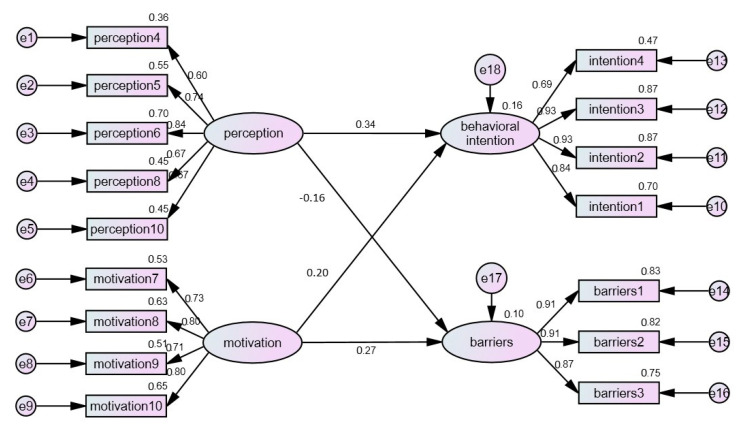
Model of behavioral intention and influencing factors for medical staff to get vaccinated against COVID-19.

**Table 1 healthcare-10-00628-t001:** Subject characteristics.

Variable	Category	Times	%	Accumulative %
Gender	Male	131	61.8	61.8
	Female	81	38.2	100.0
Age	Under 30 years old	51	24.1	24.1
	31–40 years old	70	33.0	57.1
	41–50 years old	55	25.9	83.0
	51–60 years old	29	13.7	96.7
	Above 61 years old	7	3.3	100.0
Job Title	Doctor	12	5.7	5.7
	Pharmacist	2	0.9	6.6
	Nurse	40	18.9	25.5
	Administrative staff	16	7.5	33.0
	Medical assistant	4	1.9	34.9
	Caregiver	18	8.5	43.4
	Volunteer	5	2.4	45.8
	Trainee nurse	3	1.4	47.2
	Fire rescue technician	112	52.8	100.0
Medicine category	Traditional Chinese medicine	21	9.9	9.9
	Western medicine	191	90.1	100.0
Educational Level	Senior/vocational high school or below	17	8.0	8.0
	Junior college	42	19.8	27.8
	University	102	48.1	75.9
	Postgraduate and above	51	24.1	100.0
Marital Status	Unmarried	81	38.7	38.7
	Married	121	57.1	95.8
	other	9	4.2	100.0
Current working department	Department of family medicine	2	0.9	0.9
	Department of internal medicine	9	4.2	5.2
	Department of surgery	6	2.8	8.0
	Pediatrics	3	1.4	9.4
	Department of obstetrics and gynecology	5	2.4	11.8
	Department of orthopedics	5	2.4	14.2
	Department of Department of neurosurgery	4	1.9	16.0
	Department of urology	2	0.9	17.0
	Department of ENT	1	0.5	17.5
	Department of ophthalmology	2	0.9	18.4
	Department of neurology	4	1.9	20.3
	Department of Psychiatry	4	1.9	22.2
	Department of rehabilitation	4	1.9	24.1
	Department of radiation oncology	2	0.9	25.0
	Department of clinical pathology	2	0.9	25.9
	Department of emergency medicine	5	2.4	28.3
	Department of critical care medicine	2	0.9	29.2
	Department of public health	5	2.4	31.6
	other	33	15.6	47.2
	Fire department/team	111	52.4	99.5
	Emergency medical services	1	0.5	100.0
Types of vaccines you have received (check more than one if necessary)	Influenza B vaccine	83		
	Seasonal flu vaccine	147		
	measles, mumps, German measles vaccine	119		
	Chickenpox vaccine	105		
	Diphtheria, tetanus, Pertussis vaccines	94		
	Other	26		

**Table 2 healthcare-10-00628-t002:** Comparison of the differences in behavioral intention and barriers between the medical staff of different genders.

Variable	Gender	Average Mean	Standard Deviation	T Value	*p*-Value
Behavioral intention	Male	3.69	0.63	0.88	0.37
	Female	3.61	0.69		
Barriers	Male	3.22	1.07	−2.38	0.01 *
	Female	3.54	0.85		

* *p* < 0.05.

**Table 3 healthcare-10-00628-t003:** Perception–verification analysis.

Construct	Index	Standardized Loading	Unstandardized Loading	S.E.	C.R. (t-Value)	*p*	SMC	C.R.	AVE
Perception	Perception 4	0.60	1.00				0.36	0.83	0.50
	Perception 5	0.76	1.23	0.15	8.30	***	0.58		
	Perception 6	0.82	1.13	0.14	8.31	***	0.68		
	Perception 8	0.68	0.84	0.11	7.47	***	0.46		
	Perception 10	0.67	1.08	0.15	7.42	***	0.44		

*** *p* < 0.001.

**Table 4 healthcare-10-00628-t004:** Motivation–verification analysis.

Construct	Index	Standardized Loading	Unstandardized Loading	S.E.	C.R. (t-Value)	*p*	SMC	C.R.	AVE
Motivation	Motivation 7	0.73	1.00				0.53	0.84	0.57
	Motivation 8	0.79	1.17	0.11	10.31	***	0.62		
	Motivation 9	0.71	1.08	0.12	9.36	***	0.50		
	Motivation 10	0.81	1.20	0.12	10.44	***	0.66		

*** *p* < 0.001.

**Table 5 healthcare-10-00628-t005:** Behavioral intention–verification analysis.

Construct	Index	Standardized Loading	Unstandardized Loading	S.E.	C.R. (t-Value)	*p*	SMC	C.R.	AVE
Behavioral intention	Intention 1	0.84	1.00				0.70	0.91	0.72
	Intention 2	0.93	1.06	0.06	18.10	***	0.87		
	Intention 3	0.93	1.05	0.06	17.73	***	0.86		
	Intention 4	0.68	0.71	0.06	11.21	***	0.47		

*** *p* < 0.001.

**Table 6 healthcare-10-00628-t006:** Barriers–verification analysis.

Construct	Index	Standardized Loading	Unstandardized Loading	S.E.	C.R. (t-Value)	*p*	SMC	C.R.	AVE
Barrier	Barrier 1	0.91	1.00				0.83	0.92	0.80
	Barrier 2	0.91	1.00	0.05	19.30	***	0.82		
	Barrier 3	0.87	0.97	0.05	17.95	***	0.75		

*** *p* < 0.001.

**Table 7 healthcare-10-00628-t007:** Analysis of the overall model fit.

Fit Index	Allowable Range	Correction Mode	Model Fit Judgment
χ^2^ (Chi-square)	The smaller, the better	119.25	
Ratio of χ^2^ to degrees of freedom	<3	1.19	Pass
GFI	>0.80	0.93	Pass
AGFI	>0.80	0.91	Pass
RMSEA	<0.08	0.03	Pass
CFI	>0.90	0.99	Pass
PCFI	>0.50	0.82	Pass

**Table 8 healthcare-10-00628-t008:** Empirical results of the research hypotheses.

Hypothesis	Path Relationship	Path Value	Hypothesis Supported
H1	Perception → Behavioral Intention	0.34 *	supported
H2	Perception → Barrier	−0.16 *	supported
H3	Motivation → Behavioral Intention	0.20 *	supported
H4	Motivation → Barrier	0.27 *	supported

Source: Compiled by this study. * *p* < 0.05.

## Data Availability

No specific data were used to support this study.

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
