# Peer review of "The Behavioral Intention and Influencing Factors of Medical Staff toward COVID-19 Vaccinations"

_healthcare, 2022, doi:10.3390/healthcare10040628_

Round 1

Reviewer 1 Report

The article represents a valuable research.

The abstract  is well construct .

Introduction: highlight why the study is needed, what are the contrasts that it aims to clear out; within introduction, after having problematized the research goals and gap, after present the research question define briefly the methodology: research design, data collection, and data analysis; show, in brief, the results; define how the results advance literature.

The research implications to practice and theory need to be addressed.

Also specify the limitations and future research,

Author Response

Dear Reviewer,

Thank you for the constructive suggestions and comments on our manuscript (ID: healthcare-1616088). The suggestions and comments are helpful for improving the manuscript. We are submitting the revised version of the manuscript with our responses to the suggestions and comments by the reviewer. Many thanks for your guidance.

Our responses to each suggestion and comment are as follows, and they are presented in blue texts with a grey background color in the revised manuscript.

Reviewer 2 Report

Dear Authors,

The manuscript entitled "The Behavioral Intention and Influencing Factors of Medical Staff for COVID-19 Vaccinations" is a very interesting and pertinent topic.

Please consider the following suggestions to improve your manuscript:

I suggest including the term “vaccination” in the keywords.
I suggest a justification of the statements made in lines: 36, 37, 69, 71, 95, 107 and 159.
Define the abbreviation WHO (Line 96).
When was the data collected?
Are there statistically significant differences between the sexes?
Was this study approved by an ethics committee? What? and when?
You can present an informed consent model used.

Did the study have limitations? I suggest including some.

Author Response

(The authors gave the same response as above.)
